# New Role of Water in Transketolase Catalysis

**DOI:** 10.3390/ijms24032068

**Published:** 2023-01-20

**Authors:** Olga N. Solovjeva

**Affiliations:** Belozersky Institute of Physico-Chemical Biology, Moscow State University, 119234 Moscow, Russia; soloveva_o@list.ru

**Keywords:** transketolase, thiamine diphosphate-dependent enzymes, thiamine catalysis, one-substrate reaction, mass spectrometry

## Abstract

Transketolase catalyzes the interconversion of keto and aldo sugars. Its coenzyme is thiamine diphosphate. The binding of keto sugar with thiamine diphosphate is possible only after C2 deprotonation of its thiazole ring. It is believed that deprotonation occurs due to the direct transfer of a proton to the amino group of its aminopyrimidine ring. Using mass spectrometry, it is shown that a water molecule is directly involved in the deprotonation process. After the binding of thiamine diphosphate with transketolase and its subsequent cleavage, a thiamine diphosphate molecule is formed with a mass increased by one oxygen molecule. After fragmentation, a thiamine diphosphate molecule is formed with a mass reduced by one and two hydrogen atoms, that is, HO and H_2_O are split off. Based on these data, it is assumed that after the formation of holotransketolase, water is covalently bound to thiamine diphosphate, and carbanion is formed as a result of its elimination. This may be a common mechanism for other thiamine enzymes. The participation of a water molecule in the catalysis of the one-substrate transketolase reaction and a possible reason for the effect of the acceptor substrate on the affinity of the donor substrate for active sites are also shown.

## 1. Introduction

Transketolase (TK; EC 2.2.1.1) from baker’s yeast *Saccharomyces cerevisiae* is a homodimer with structurally identical subunits [1,2] and has two initially identical active sites with the same catalytic activity [3]. The binding and cleavage of substrates occur on the coenzyme thiamine diphosphate (ThDP), bound both directly to TK and via a divalent cation [4]. ThDP binding does not occur in the absence of a cation in the active site. In native TK, Ca^2+^ was found [5]. Two active sites are localized at the interface of the subunits, and the binding of each ThDP molecule to apoTK includes amino acid residues of both subunits [4]. This is the reason that the active sites have functional non-equivalence in the binding of the cofactors Ca^2+^ [6] and ThDP [3,7], which is induced by cofactor binding in one (conditionally first) active site.

TK catalyzes the interconversion of keto sugars and aldo sugars. It catalyzes two processes: C-C bond cleavage in ketose (donor substrate) and the reversible transfer of the resulting two-carbon fragment to aldose (acceptor substrate) [8]. TK is characterized by broad substrate specificity [9]. The donor substrates are xylulose 5-phosphate (X5P), fructose 6-phosphate (F6P), sedoheptulose 7-phosphate, hydroxypyruvate (HPA), erythrulose, etc.; the acceptor substrates are ribose 5-phosphate (R5P), erythrose 4-phosphate, glyceraldehyde 3-phosphate (G3P), etc. In addition to the two-substrate (transferase) reaction, TK catalyzes the one-substrate reaction with only a donor substrate [10,11]. In the presence of Ca^2+^ in holoTK, two K_m_ values are determined for the binding of donor substrates in two-substrate [12,13,14] and one-substrate [6,10,11,15] reactions. The first stage of both reactions is the same—binding keto sugar and removing aldo sugar from it, which is shortened by two carbon atoms. Glycolaldehyde (GlyA) remains bound to the C2 of the thiazole ring of ThDP [2]. Thereafter, in the two-substrate reaction, another aldo sugar is bound with GlyA, and a new keto sugar is formed, elongated by two carbon atoms, and is then released into the medium [2]. In the one-substrate reaction, the first molecule of the formed GlyA is transferred from the thiazole ring to the aminopyrimidine ring of ThDP. Then, binding and cleavage of the second keto substrate occurs, after which the two GlyA molecules condense into erythrulose, which is released into the medium [15,16].

To bind the first keto substrate molecule, it is necessary to deprotonate the C2 of the ThDP thiazole ring. According to crystallographic data, this is due to the fact that the amino group of the aminopyrimidine ring directly attaches the proton [11,17] (Figure 1A). This leads to the formation of an imino tautomeric form of ThDP (compound **3** in Figure 1A), followed by the deprotonation of C2 (compounds **5**, **6** in Figure 1A). For TK from baker’s yeast and *E. coli*, Hecquet showed, using a quantum mechanical/molecular mechanical method, that the C2 deprotonation reaction can be carried out with additional participation of a water molecule (Figure 1B) [18]. According to this, on the contrary, the formation of the imino tautomeric form of ThDP may be a consequence of the decarboxylation of C2 (compound **7** in Figure 1B). The scheme of Hecquet complements and does not contradict the conventional scheme.

The role of water in the functioning of enzymes is diverse and interesting to study. It has previously been shown that proton transfer can be carried out sequentially through several water molecules [19]. For ThDP-dependent enzymes, it has been shown that the reciprocal influence of active sites located in the cleft between identical subunits occurs using bridges of several water molecules [20,21]. “In human transketolase, the putative proton wire includes six glutamates in total—three provided by each subunit—and several water molecules; together these link the two active sites of the dimer over a distance of around 25 Å” [22]. In [23], it was shown that all three nitrogen atoms of the ThDP aminopyrimidine ring are capable of forming a hydrogen bond with a water molecule. The inclusion of two water molecules to simulate the apoenzymatic medium reduces the activation energies of both direct and water-mediated ylide generation.

The protonation of ThDP enamine by the Glu residue of the active site through an intermediate water molecule was also proposed by Lobell and Crout for yeast pyruvate decarboxylase based on their molecular mechanics calculations [24], and by Sergienko and Jordan based on their study of site-directed mutagenesis [25], as well as for pyruvate decarboxylase from *Zymomonas mobilis* [26]. The direct role of the water molecule in catalysis has also been shown for other enzymes [27,28,29,30,31,32,33,34].

In this work, using mass spectrometry, it was shown that after binding to TK, ThDP covalently binds to the water molecule. The one-substrate TK reaction also requires the participation of water.

## 2. Results and Discussion

### 2.1. ThDP-O Formation and Cyanoborohydride Reduction of ThDP and ThDP-O

We performed mass spectrometry (MS) analysis of ThDP and its intermediates, formed after the incubation of holoTK with various substrates. For the commercial ThDP, only *m*/*z* 425.045 is detected using MS. After the binding of ThDP to TK and its subsequent separation via protein denaturation, along with *m*/*z* 425.045, *m*/*z* 441.040 is determined (Figure 2), which is more than the *m*/*z* of ThDP per one oxygen atom (ThDP-O). Table 1 and Figure 2 show the amplitudes of the formed ThDP-O relative to the amplitudes of ThDP in samples prepared without the addition of substrates and in the presence of several substrates—donors and acceptors. In the absence of substrates, the amplitude of ThDP-O is 0.4%; in the presence of donor substrates, it is 4–44%. Thus, the addition of donor substrates significantly increases the amount of formed ThDP-O, which is an indication of the need for its formation for catalysis. When incubating holoTK in the presence of NaBH_3_CN, the amplitudes of the formed ThDP-O decrease and are 2–6% upon incubation of holoTK in the presence of donor and acceptor substrates. So, it is not possible to draw any conclusions regarding the effect of acceptor substrates.

In the presence of NaBH_3_CN, ThDP (*m*/*z* 425.045) is partially reduced to dihydrothiamine diphosphate (DHThDP) (*m*/*z* 427.061) and tetrahydrothiamine diphosphate (THThDP) (*m*/*z* 429.076), and ThDP-O (*m*/*z* 441.040) is partially reduced to DHThDP-O (*m*/*z* 443.056) and THThDP-O (*m*/*z* 445.071) (Table 1, Figure 2). THThDP was previously prepared from ThDP via reduction with NaBH_4_ [35,36]. The thiazole ring is shown to be tetrahydrated in ThDP [37]. In the presence of NaBH_3_CN, in addition to THThDP, DHThDP is formed. During their fragmentation, when the molecule is broken in half, the *m*/*z* of the aminopyrimidine ring remains at 122.072, and the *m*/*z* of the thiazole ring is 303.981 for ThDP, 305.997 for DHThDP, and 308.012 for THThDP. Therefore, in the presence of NaBH_3_CN, the thiazole ring of ThDP is also dehydrogenated and tetrahydrated.

The total amplitudes of the formed DHThDP and THThDP relative to the amplitude of ThDP are small in the absence of substrates (6%) and in the presence of the donor substrates HPA (13%) and F6P (18%). In the presence of the acceptor substrates G3P and R5P, the total amplitudes of DHThDP and THThDP increase significantly and are equal, respectively, to 500 and 900% of the amplitude of ThDP. GlyA, which is both a donor substrate and an acceptor substrate [38], occupies an intermediate position. For it, the total amplitudes of DHThDP and THThDP are 150% (Table 1). As previously shown, the incubation of holoTK with an acceptor substrate in the presence of NaBH_3_CN results in its irreversible inhibition. The enzyme activity is restored after displacement of the formed intermediate by adding ThDP to the incubation medium. Thus, inhibition is caused by the loss of ThDP catalytic function [39]. Mass spectrometric analysis shows that, under these conditions, no derivatives of ThDP are detected, which would indicate the binding of an acceptor substrate to it, but a significant increase in the amount of formed DHThDP and THThDP is found. Therefore, it can be argued that the inhibition of TK by NaBH_3_CN in the presence of an acceptor substrate is only the result of the reduction of ThDP to DHThDP and THThDP. This fact is consistent with data from X-ray crystallography and of isothermal titration calorimetry, showing that in the absence of donor substrates or their derived intermediates, R5P binds directly to TK, and not to ThDP [40]. The significant increase in the DHThDP and THThDP formed in the presence of the acceptor substrate can be explained by the fact that the binding of the acceptor substrate prevents the formation of the V-conformation of ThDP (in which the ThDP rings draw near), and thus, facilitates the access of the reducing agent to the ThDP thiazole ring. This question remains to be clarified. According to the crystallography of human TK with sub-angström resolution, the “enzyme-promoted destabilization of intrinsically stable intermediates appears to be a general principle of enzyme catalysis, in addition to transition-state stabilization and substrate destabilization” [41].

### 2.2. The Reason for the Influence of the Acceptor Substrate on the Kinetic Parameters for the Donor Substrate

If we compare the kinetic parameters for donor substrates in the absence and in the presence of the acceptor substrate R5P, it can be seen that in the presence of R5P, both values of K_m_ increase (Table 2). The exception is GlyA, which is both a donor substrate and an acceptor substrate [38] and, accordingly, the reaction with it alone is essentially two-substrate. For typical donor substrates, such a decrease in its affinity in the presence of an acceptor substrate can be due to two reasons—(i) an unfavorable change in the structure of the active site for binding the donor substrate as a result of binding the acceptor substrate, or (ii) competitive binding of the acceptor and the donor substrate. To select one of these two possibilities, we determined the affinity of X5P for the active sites of holoTK at several fixed R5P concentrations (Figure 3 and Table 3). In the case of the competitive interaction of substrates, a gradual increase in K_m_ could be expected for X5P with an increase in R5P concentration. As in the one-substrate reaction with different substrates [6,15], two K_m_ values are determined in the two-substrate reaction and a sharp drop in activity is observed when a certain concentration of X5P is reached (Figure 3). The reason seems to be the same—switching of the alternate binding of X5P in both active sites to its simultaneous binding, as shown in article [6]. The kinetic parameters were determined by independently fitting both parts of the curves (inserts in Figure 3).

The reaction mixture (1 mL) contained 50 mM glycylglycine, pH 7.6, 0.1 mM Ca^2+^, 0.1 mM ThDP, 7 mM sodium arsenate, 3.2 mM dithiothreitol, 0.3 mM NAD^+^, 3 U GAPDH, 0.187 μg TK, and (1) 150 μM, (2) 300 μM, (3) 450 μM, and (4) 1500 μM R5P. The reaction was started by adding holoTK.

Previously, the inhibitory concentration of R5P was shown to be above 500 μM [42]. K_m_1 for R5P was equal to 60 μM and K_m_2-250 μM [12]. We took three non-inhibitory concentrations of R5P (150, 300, and 450 μM) and one inhibitory (1500 μM). K_m_1 and K_m_2 for X5P were the same in all cases. Only K_m_2 at the lowest R5P concentration of 150 μM (concentration below its K_m_) was twice reduced. It was concluded that an increase in R5P concentration does not affect the affinity of X5P for the enzyme, which means that there is no competition between the donor substrate and the acceptor substrate for binding to the TK active sites. Consequently, the effect of the acceptor substrate is due only to the fact of its binding with residues of the active site of holoTK, which, apparently, changes the interposition of the coenzyme rings. The calculation of K_m_ was carried out according to the same formula as for the one-substrate reactions [6,15], that is, assuming that at a low concentration of X5P, both active sites have the same affinity for the substrate and work simultaneously, and when the concentration of X5P increases to a certain value, the affinity for the substrate of each of them decreases sharply to the same extent due to the fact that the active sites begin to work alternately.

The changes in V2 at different R5P concentrations (Table 3) coincided with those obtained in [42]—V2 increases as the R5P concentration approaches saturation, and then, decreases with a further increase in R5P concentration. For V1, there was neither an initial increase nor a subsequent decrease with an increase in R5P concentration (Table 3).

### 2.3. The Structure of ThDP-O: Proposed Participation of Water in Activation of ThDP

Figure 4 shows the results of MS/MS ThDP [16] and ThDP-O. Typical ThDP fragmentation (*m*/*z* 425.045) involves the cleavage of monophosphate (*m*/*z* 345.079 and 327.068) and diphosphate (*m*/*z* 249.118 and 263.096), breaking the molecule in half (*m*/*z* 122.071 + 303.981), as well as the formation of rearrangement ions (*m*/*z* 220.049, 202.038 + 224.015) [16] (Figure 4A and Figure 5A, Table 4). When the ThDP molecule is broken in half, the amplitude of mass 122.072 (aminopyrimidine ring) is about 20 times smaller than the amplitude of mass 303.981 (diphosphothiazole), since the aminopyrimidine ring can be negatively charged to a small extent. Obviously, the quantitative ratio of both formed rings is the same. Such a difference in amplitudes makes it possible to determine which of the two rings a certain mass of any fragment belongs to.

The fragmentation of ThDP-O (*m*/*z* 441.040) also results in the cleavage of monophosphate (*m*/*z* 361.074 and 343.063) and diphosphate (*m*/*z* 265.112 and 247.102), breaking the molecule in half (*m*/*z* 122.071 + 303.981), and the formation of rearrangement ions (*m*/*z* 220.049, 202.038) (Figure 4B and Figure 5B). For our conclusions, fragments with *m*/*z* 424.037 and 423.029 are important, which differ from the *m*/*z* of ThDP in one- and two-hydrogen atoms. Accordingly, during fragmentation, ThDP-O is cleaved not by O, but by HO (*m*/*z* 424.037) and H_2_O (*m*/*z* 423.029). This proves that water is covalently bound to ThDP. Unlike ThDP-O fragmentation, in the fragmentation of ThDP and its intermediates with substrates, only *m*/*z* 425 was detected, not 423 and 424 [15,16]. The fact that *m*/*z* 441 is absent from ThDP prior to its binding to TK confirms that ThDP-O is enzymatically formed. Further evidence that the water molecule in ThDP-O binds to both ThDP rings is its increased strength. *m*/*z* 441 is difficult to fragment. Only monophosphate (intermediate with *m*/*z* 361.074) is easily cleaved off (Figure 5B, Table 4).

Based on the data obtained, it can be assumed that deprotonation of the C2 thiazole ring of ThDP occurs as a result of the elimination of a water molecule previously bound to ThDP. It is especially important that the binding of the water molecule to ThDP can occur only with the participation of the enzyme. These results confirm the Hecquet data obtained using the quantum mechanical/molecular mechanical method, according to which the deprotonation of C2 on the thiazole ring occurs with the participation of a water molecule [18]. The Hecquet scheme assumes that during ThDP deprotonation, water remains bound to His103 and His481 (Figure 1B), and the C2 of ThDP is deprotonated not by ThDP N4’, as stated earlier, but by His481, involving a bridge of the water molecule (H2O688A). We additionally show via MS/MS analysis that the deprotonation of C2 occurs due to the fact that water covalently binds to ThDP.

It is for this reason that the deprotonation of ThDP-O occurs when a water molecule is cleaved from ThDP-O.

Since the fragmentation of ThDP-O (*m*/*z* 441) leads to the formation of fragments without both one hydrogen atom (*m*/*z* 424) and two hydrogen atoms (*m*/*z* 423), it can be assumed that the deprotonation of ThDP can occur in two stages.

However, it cannot be ruled out that water remains bound to ThDP, and the donor substrate displaces the bound water molecule in the active site, as has been shown, for example, for D-psicose 3-epimerase [43].

For the ThDP analog 4’-methylamino-ThDP, it was shown that when it is used as a coenzyme and a donor substrate is added, only the first act of catalysis occurs—the formation of dihydroxyethylthiamine diphosphate (DHEThDP). The rate of DHEThDP formation in the case of native and methylated ThDP was similar [44]. Thus, it can be concluded that the absence of a native amino group in the aminopyrimidine ring of ThDP does not affect the deprotonation rate of C2 ThDP, which is a necessary condition for the first act of catalysis. This is an indirect confirmation of the mediated role of the amino group in carbanion formation.

For holoTK, ThDP-O amplitude in MS is 0.4% of ThDP amplitude. When incubating holoTK with F6P, HPA, and GlyA, the amplitude of ThDP-O increases to 4, 12, and 44%, respectively (Table 1). Thus, the presence of donor substrates contributes to an increase in the amount of deprotonated ThDP bound to the water molecule required for catalysis. Donor substrates have previously been shown to increase the affinity of the coenzyme for TK [45], possibly for this reason. The binding of ThDP to TK occurs in two stages: first, a catalytically inactive complex TK–ThDP is formed, which is then transformed into a catalytically active complex [46]. It can be assumed that in the catalytically active complex, the coenzyme is in the form of ThDP-O. It is the imino form of ThDP that is active [11,47,48]. The formation of the imino form probably follows the binding of ThDP to water.

### 2.4. The Role of Water in the One-Substrate Transketolase Reaction

In the first stage of the one-substrate reaction, the first keto substrate binds, the aldose shortened by two carbon atoms is cleaved, and DHEThDP is formed, which is GlyA bound to ThDP at the C2 of its thiazole ring. The GlyA residue formed as a result of cleavage of the first keto substrate is then transferred from the C2 of the thiazole ring of ThDP to the NH_2_ of its aminopyrimidine ring [15,16]. At the second stage of the one-substrate reaction, binding and cleavage of the second keto substrate also occur at the C2 of the thiazole ring of ThDP [15]. The transfer of the first GlyA residue to the aminopyrimidine ring proceeds through the water cleavage step. As a result, GlyA is bound to both rings, after which water is added and the bond of GlyA with the thiazole ring cleaved (Figure 6). This stage is rate-limiting. Due to this, the rate of the one-substrate reaction is 50 times lower than the rate of the two-substrate reaction [10,11].

In the second stage of the one-substrate reaction, the second molecule of the donor substrate binds and cleaves off and erythrulose is formed from two GlyA molecules. Both reactions take place on the thiazole ring of ThDP [15]. On the MS, two *m*/*z*—527 and 543 are determined, differing in one oxygen. Figure 7 shows the half fragments obtained by MS/MS of this *m*/*z*. For both *m*/*z*, two types of molecules are observed—in the first two GlyA are linked to both ThDP rings, in the second erythrulose is linked to the thiazole ring of ThDP. The transition from the first type to the second appears to be due to water loss (Figure 8).

## 3. Materials and Methods

### 3.1. Materials

ThDP, glycylglycine, NAD^+^, GlyA, HPA, F6P, R5P, G3P, CaCl_2_, NaCNBH_3_, and formic acid were obtained from Sigma Aldrich Chemie GmbH (Schnelldorf, Germany), trichloroacetic acid from MP Biomedicals (Eschwege, Germany), and Sephadex G-50 from Pharmacia (Stockholm, Sweden). Other reactants were of extra-pure grade.

### 3.2. Transketolase Purification

Transketolase was isolated from baker’s yeast *Saccharomyces cerevisiae* on an immunoaffinity column as described earlier [49], aliquoted, and stored frozen in 20 mM potassium phosphate buffer with 0.3 M ammonium sulphate, pH 7.6. The enzyme was homogeneous by SDS-PAGE. The concentration of TK was determined spectrophotometrically using A^1%^_1cm_ = 14.5 at 280 nm [50]. Prior to use, the TK solution was passed through a Sephadex G-50 column, equilibrated with 50 mM glycylglycine, pH 7.6.

### 3.3. Determination of Thiamine Diphosphate Concentration

The concentration of ThDP was determined spectrophotometrically by measuring the optical density at 272.5 nm (using the molar extinction coefficient 7500 M^−1^cm^−1^) [51].

### 3.4. Preparation of Holotransketolase

TK at 10–15 mg/mL was supplemented with 2.5 mM CaCl_2_ and ThDP, whose concentration was twice the molar concentration of TK (i.e., one ThDP for each active site).

### 3.5. Preparation of Xylulose 5-Phosphate

X5P was obtained enzymatically using a transketolase reaction with HPA and G3P [52]. Its concentration was measured with excess of R5P using a transferase transketolase reaction.

### 3.6. TK Activity Measurement

The catalytic activity of TK in the transferase reaction was measured by the rate of NAD^+^ reduction in a coupled system with glyceraldehyde 3-phosphate dehydrogenase at 25 °C [53]. The reaction mixture in the final volume of 1 mL contained 50 mM glycylglycine, pH 7.6, 7 mM sodium arsenate, 3.2 mM dithiothreitol, 0.1 mM CaCl_2_, 0.1 mM ThDP, 0.3 mM NAD^+^, 3 U glyceraldehyde 3-phosphate dehydrogenase, 0.5 mM X5P, and 1.5 mM R5P. The reaction was initiated via the addition of 0.2 mkg/mL holoTK. The specific activity of TK was 50 U/mg.

### 3.7. Determination of the Concentration of X5P and R5P

The concentration of X5P was measured in the transferase reaction with 2 mkg/mL holoTK and 1.5 mM R5P. The concentration of R5P was measured in the transferase reaction with 2 mkg/mL holoTK and 0.5 mM X5P. Since in [15], we showed that in the presence of both substrates, the one-substrate transketolase reaction also occurs, the concentration of R5P was determined in a two-beam mode using a reference cuvette that did not contain R5P. The reaction was started with holoTK by adding it simultaneously to both cuvettes. The observed linear dependence of the resulting change in optical density on the amount of added X5P showed that the rate of a one-substrate reaction in the presence of R5P is the same as in its absence. The amount of possible X5P in the R5P solution was determined without adding X5P and was taken into account in the calculations.

### 3.8. Determination of K_m_ for X5P

The reaction mixture at the K_m_ measurement contained 50 mM glycylglycine, pH 7.6, 0.1 mM Ca^2+^, 0.1 mM ThDP, 0.187 mkg/mL TK, 7 mM sodium arsenate, 3.2 mM dithiothreitol, 0.3 mM NAD^+^, 3 U GAPDH, fixed concentrations of R5P and 7–700 mkM X5P in 1 mL. The reaction was measured spectrophotometrically at 340 nm and started by adding holoTK.

The K_m_ values were determined by fitting the curve of the experimental dependence of the reaction rate to the substrate concentration. Each point on the experimental curve is the initial reaction rate when adding different concentrations of the substrate. The experimental data could only be analyzed using the Michaelis equation v = V·[S]/([S] + K_m_), as we proved in [6,15] for a one-substrate reaction with some substrates. The sharp decrease in enzyme activity cannot be described by the formula for interacting active sites E ↔ ES1 ↔ ES2 [6].

### 3.9. Preparation of Samples for Mass Spectrometry

HoloTK was incubated with 10 mM HPA, 5 mM GlyA, or 10 mM F6P for 4 h at room temperature in the presence or absence of 10 mM NaCNBH_3_. Following incubation, the sample was passed through a Sephadex G-50 column equilibrated with 30 mM NH_4_OH and 40 mM formic acid, pH 6.5. The fractions containing protein with bound ThDP and ThDP derivatives (the intermediates) were collected. The protein was then denatured with 0.6% trichloroacetic acid and frozen for 24 h, and the residue was removed via centrifugation. The supernatant was used for mass spectrometry.

### 3.10. ESI-MS and MS/MS Analysis

The samples were in 30 mM NH_4_OH, 40 mM formic acid, and 0.6% trichloroacetic acid at pH 4.0, dissolved in water.

The samples were analyzed using an LTQ Orbitrap mass spectrometer. Each sample was injected at 3 mkL/min with a nebulizer gas flow of 8 L/min. The temperature of the inlet capillary was 270 °C, and the capillary voltage, 3.6 kV. The mass spectra in the mode of registration of negative ions were measured using an FTMS analyzer with a mass-to-charge ratio (*m*/*z*) resolution of 60,000 in the *m*/*z* range of 200–1000. The mass accuracy was above 5 ppm. Tandem mass spectra were used to clarify the composition and structure of the analyte. The appropriate ions were isolated in a window with a mass of 2 Da and subjected to collision-induced dissociation (CID) and higher-energy C trap dissociation (HCD). The resulting spectra were obtained by averaging 300 scans.

The same *m*/*z* were obtained using a 7-tesla Apex Ultra FT ICR mass spectrometer (Bruker Daltonics, Bremen, Germany). The samples were ionized via electrospray under the following conditions: spray shield voltage—3.6 kV, capillary voltage—4.1 kV, capillary temperature—200 °C, and flow rate—90 mkl/h. The full MS spectra were measured in broadband mode in a 50–1000 *m*/*z* range. *m*/*z* accuracy was above 5 ppm.

MS/MS analysis was performed only using Orbitrap because initially uncharged fragments were not visible on the Apex Ultra. All molecular ions were 1-charged.

## 4. Conclusions

We used mass spectrometry to study the catalytic mechanism of the one-substrate TK reaction. Mass spectrometry successfully supplements kinetic and structural studies. When crystals of TK from different sources are obtained with donor substrates, either DHEThDP [54,55] or only molecules of the initial substrates [40] are detected. Using mass spectrometry, we were able to determine the *m*/*z* and conduct MS/MS for all intermediates of a one-substrate transketolase reaction, after the incubation of holoTK with donor substrates, the removal of unreacted substrates and cofactors via gel filtration, and the isolation of intermediates linked to the enzyme. These were ThDP with bound GlyA with *m*/*z* 467 and 485, and with bound erythrulose (or two GlyA) with *m*/*z* 527 and 543. MS/MS analysis can determine which of the two ThDP rings the ligand is bound to [15,16]. In this article, using MS/MS analysis, we showed that covalent binding of the water molecule to ThDP occurs enzymatically, which confirms the Hecquet data [18]. Through this, apparently, the formation of the carbanion and a transition of ThDP to the active imino form occurs. The fact shown in this article that R5P facilitates access of the reducing agent to the thiazole ring of ThDP (the amount of DHThDP and THThDP formed under the action of the reducing agent increases) confirms the data of structural studies showing that R5P forms a non-covalent complex with TK [40]. Mass spectrometry turned out to be very informative when studying individual stages of the transketolase reaction. The low rate of the one-substrate reaction made it possible to identify even the reaction products associated with ThDP. For other thiamine enzymes, inactive substrate analogs appear to have to be used to solve a similar problem.

## Figures and Tables

**Figure 1 ijms-24-02068-f001:**
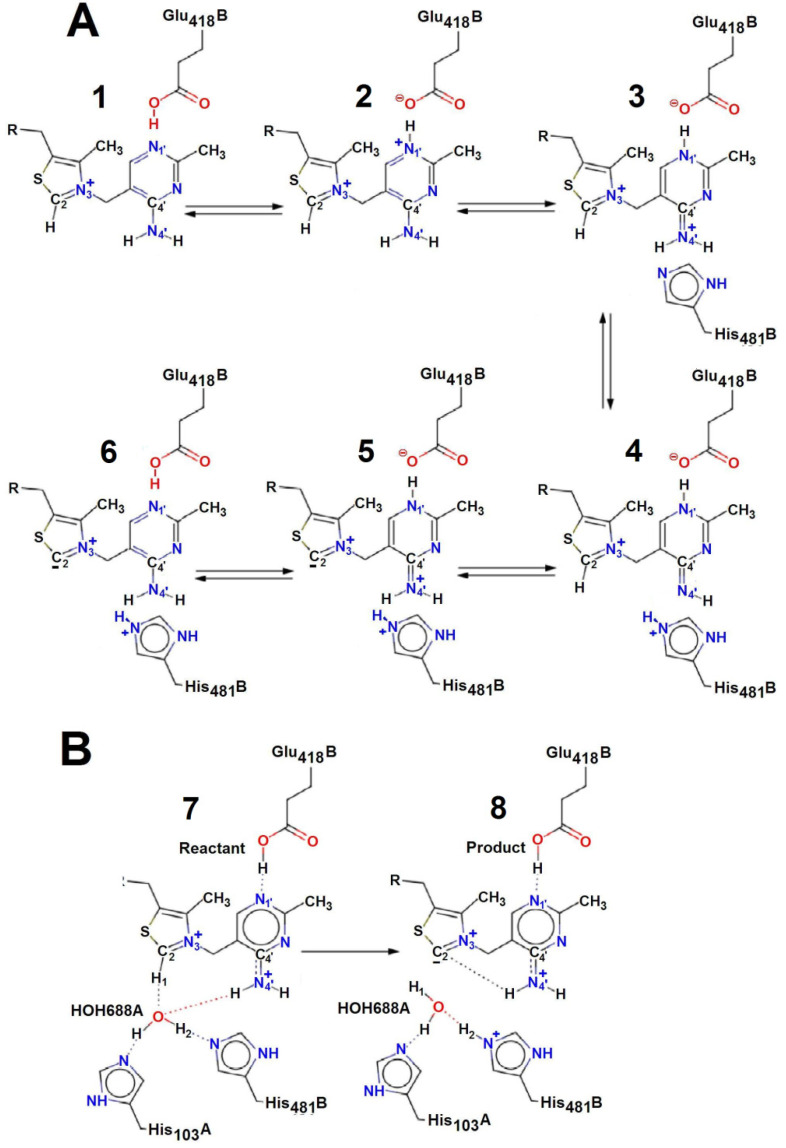
ThDP activation mechanism. (**A**)—traditional mechanism, (**B**)—mechanism with direct participation of water molecule [18].

**Figure 2 ijms-24-02068-f002:**
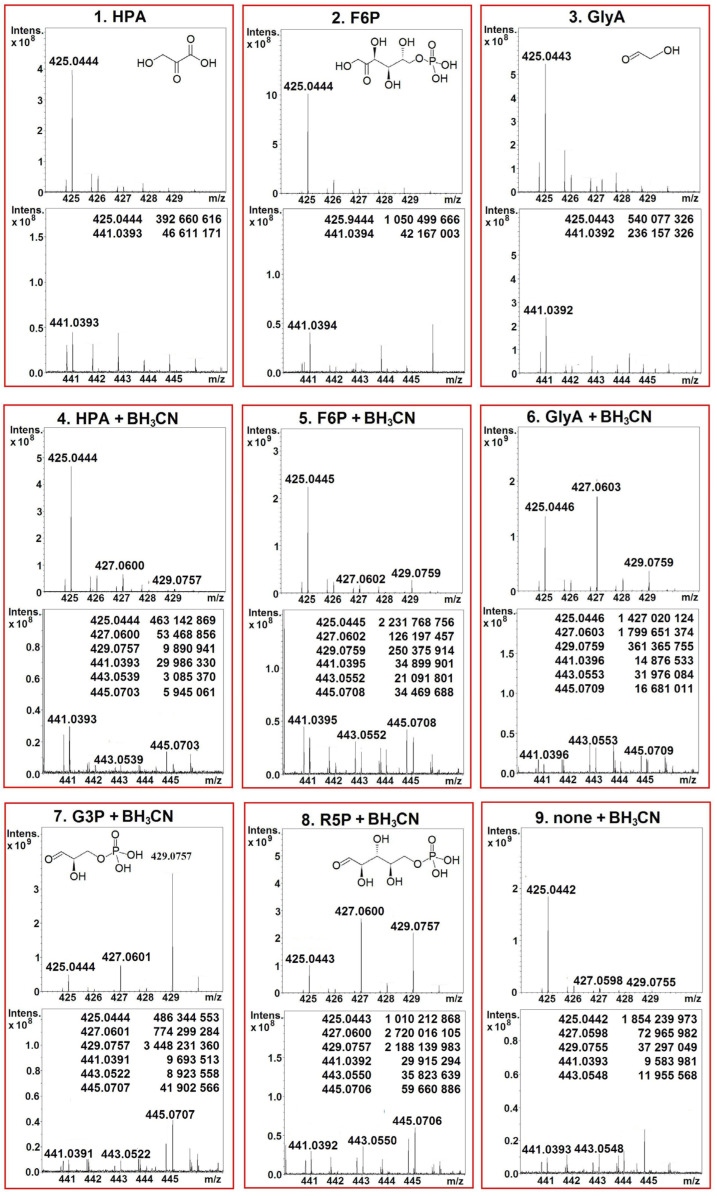
Typical MS spectra of ThDP and its intermediates after incubation of holoTK with donor substrates without (**1**–**3**) and with (**4**–**6**) NaBH_3_CN; with acceptor substrates (**7**,**8**) and without substrates (**9**) with NaBH_3_CN, at *m*/*z* intervals of 424–430 and 440–446. Inserts—absolute intensities of *m*/*z* 425 and 441 and its hydrated forms. Data from Apex Ultra.

**Figure 3 ijms-24-02068-f003:**
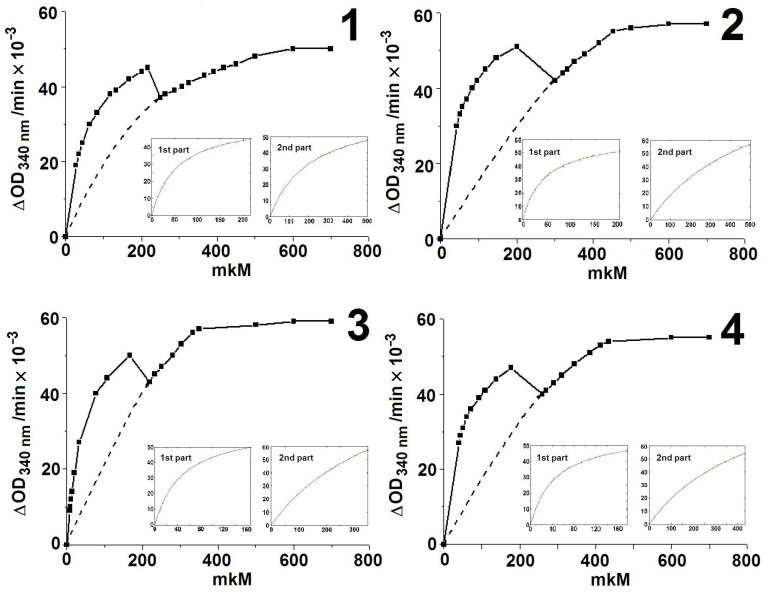
Dependence of holoTK activity in the two-substrate reaction on concentration of X5P. All curves contain data from one experiment of the same type. Inserts—fitting both parts of the curves.

**Figure 4 ijms-24-02068-f004:**
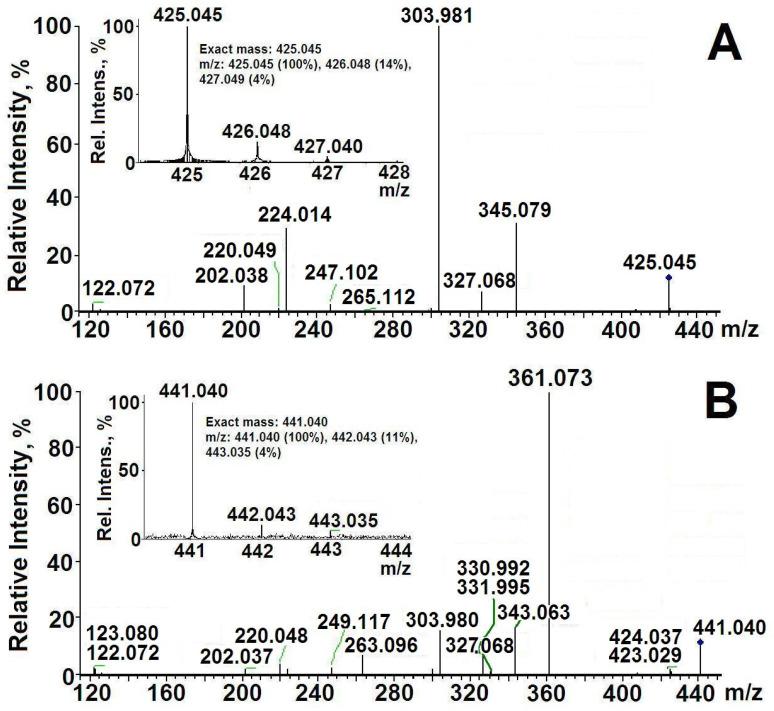
Typical MS/MS spectra of *m*/*z* 425.045 [16] (**A**) and 441.040 (**B**).

**Figure 5 ijms-24-02068-f005:**
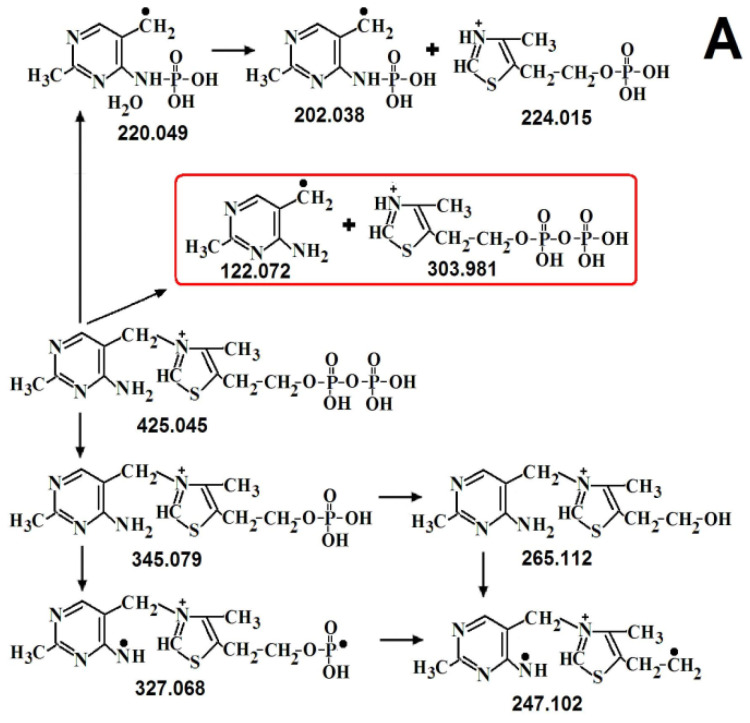
Proposed fragmentation of ThDP (**A**) [16] and ThDP-O (**B**). In red boxes are fragments formed when the molecule is broken in half.

**Figure 6 ijms-24-02068-f006:**
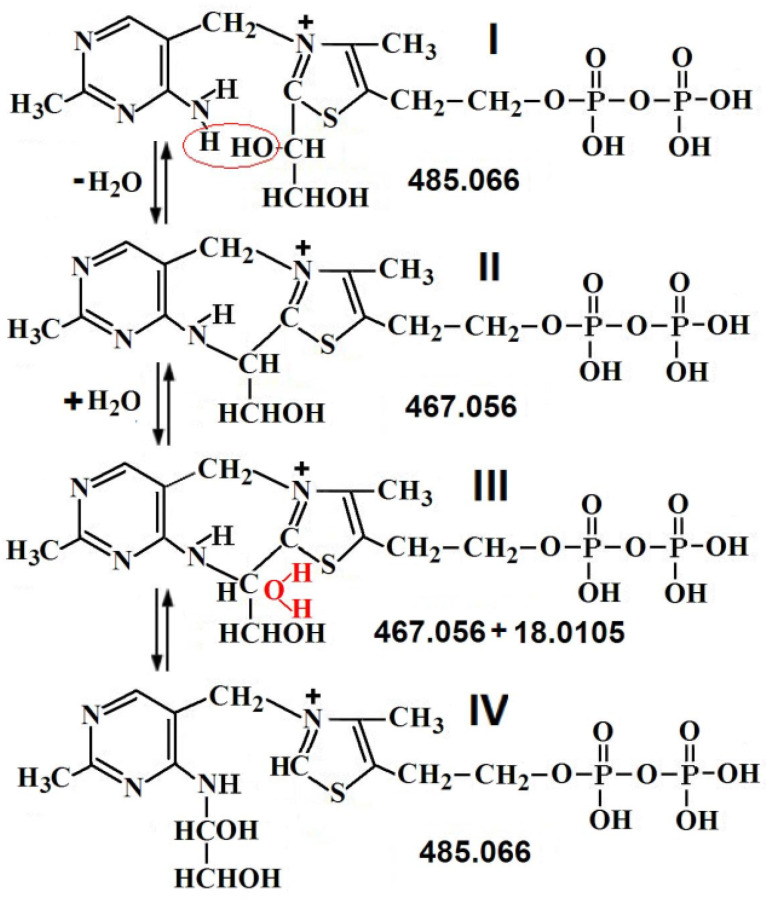
Proposed mechanism of the first stage of the one-substrate transketolase reaction.

**Figure 7 ijms-24-02068-f007:**
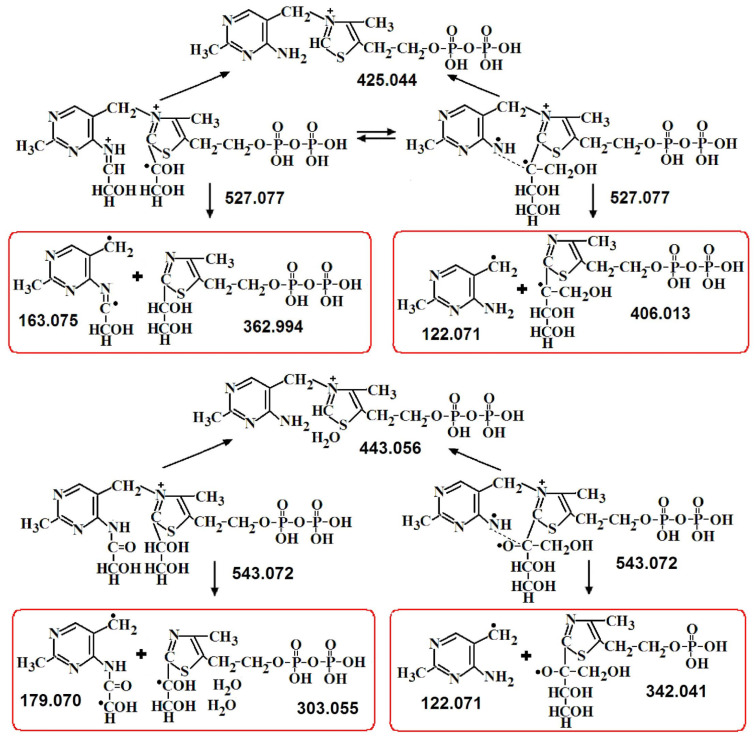
Some proposed fragments of *m*/*z* 527.060 and *m*/*z* 543.055 [15]. In red boxes are fragments formed when the molecule is broken in half.

**Figure 8 ijms-24-02068-f008:**
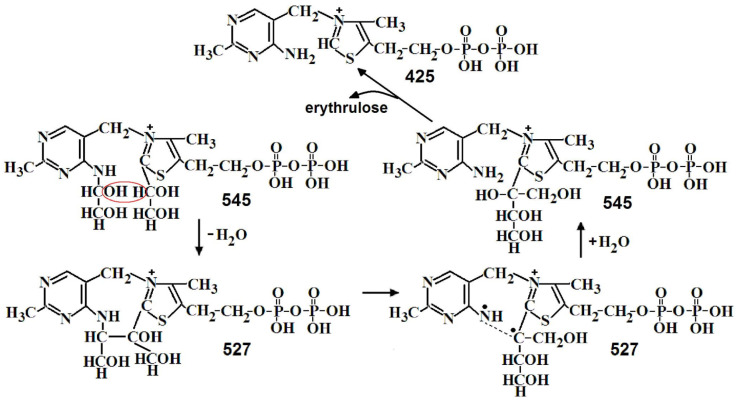
Proposed mechanism of the second stage of the one-substrate transketolase reaction.

**Table 1 ijms-24-02068-t001:** Amplitudes of ThDP intermediates relative to amplitude of ThDP in MS spectra. Data from Apex Ultra.

Substrate	With NaBH_3_CN					Without NaBH_3_CN
	*m*/*z* 427	*m*/*z* 429	*m*/*z* 441	*m*/*z* 443	*m*/*z* 445	*m*/*z* 441
none	4%	2%	0.5%	0.6%		0.4%
HPA	11%	2%	6%	0.7%	1.3%	12%
F6P	6%	12%	1.6%	1%	1.5%	4%
GlyA	125%	25%	1%	2%	1%	44%
G3P	170%	710%	2%	2%	9%	not measured
R5P	270%	216%	3%	3%	6%	not measured

**Table 2 ijms-24-02068-t002:** Kinetic parameters of holoTK in one-substrate and two-substrate reactions.

Donor Substrate	Acceptor Substrate	K_m_1 (μM)	K_m_2 (μM)	V1 (U/mg)	V2 (U/mg)
HPA [6]	none	21 ± 1.5	110 ± 5	0.032 ± 0.002	0.044 ± 0.002
HPA [6]	1.5 mM R5P	205 ± 30	475 ± 6	2.52 ± 0.29	4.08 ± 0.1
X5P [6]	none	0.39 ± 0.02	24.34 ± 1.69	0.20 ± 0.003	0.32 ± 0.01
X5P [14]	1.5 mM R5P	25 ± 6	773 ± 300	15 ± 3	69 ± 17
F6P [15]	none	0.22 ± 0.02	1.16 ± 0.2	8.1 ± 0.3 × 10^−3^	9.1 ± 0.3 × 10^−3^
F6P [15]	1.5 mM R5P	1.3 ± 0.15	8.5 ± 0.15	56 ± 3 × 10^−3^	64 ± 3 × 10^−3^
GlyA [15]	none	150 ± 20	1800 ± 300	0.19 ± 0.02 × 10^−3^	0.24 ± 0.03 × 10^−3^
GlyA [15]	1.5 mM R5P	470 ± 50	1700 ± 30	0.43 ± 0.05 × 10^−3^	0.51 ± 0.04 × 10^−3^

**Table 3 ijms-24-02068-t003:** Kinetic parameters of holoTK in two-substrate reaction for X5P binding with different concentrations of R5P.

R5P (μM)	K_m_1 (μM)	K_m_2 (μM)	V1 (U/mg)	V2 (U/mg)
150	54.1 ± 1.5	209 ± 63	47.7 ± 0.44	58.0 ± 0.65
300	45.2 ± 1.2	540 ± 53	53.4 ± 0.48	101.5 ± 5.8
450	47.7 ± 1.2	429 ± 22	55.1 ± 0.59	109.3 ± 3.3
1500	44.4 ± 1.3	482 ± 17	50.0 ± 0.53	98.2 ± 2.0

**Table 4 ijms-24-02068-t004:** MS/MS fragments of ThDP and ThDP-O.

Molecular Formula	Calculated Mass (*m*/*z*)	Experimental Mass (*m*/*z*)	Relative Intensity	Calculated Mass (*m*/*z*)	Experimental Mass (*m*/*z*)	Relative Intensity
C_12_O_8_H_19_S_1_N_4_P_2_				441.040	441.040	10.05
C_12_O_7_H_19_S_1_N_4_P_2_	425.045	425.045	11.25			
C_12_O_7_H_18_S_1_N_4_P_2_				424.037	424.037	1.32
C_12_O_7_H_17_S_1_N_4_P_2_				423.029	423.029	0.48
C_12_O_5_H_18_S_1_N_4_P_1_				361.074	361.073	100
C_12_O_4_H_18_S_1_N_4_P_1_	345.079	345.079	32.42			
C_12_O_4_H_16_S_1_N_4_P_1_				343.063	343.063	16.57
C_7_O_7_H_14_S_1_N_2_P_2_				331.999	331.995	0.07
C_7_O_7_H_13_S_1_N_2_P_2_				330.992	330.992	0.06
C_12_O_3_H_16_S_1_N_4_P_1_	327.068	327.068	7.55			
C_6_O_7_H_12_S_1_N_1_P_2_	303.981	303.981	100	303.981	303.980	0.11
C_12_O_1_H_17_S_1_N_4_	265.113	265.112	0.48			
C_12_O_1_H_15_S_1_N_4_				263.097	263.096	6.83
C_12_H_17_S_1_N_4_				249.118	249.117	0.05
C_12_H_15_S_1_N_4_	247.102	247.102	2.94			
C_6_O_4_H_11_S_1_N_1_P_1_	224.015	224.015	29.27			
C_6_O_4_H_11_N_3_P_1_	220.049	220.049	0.96	220.049	220.048	0.29
C_6_O_3_H_9_N_3_P_1_	202.038	202.038	10.4	202.038	202.037	0.11
C_6_H_9_N_3_				123.080	123.080	0.06
C_6_H_8_N_3_	122.072	122.072	3/02	122.072	122.072	0.07

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
