# Peer review of "New Role of Water in Transketolase Catalysis"

_ijms, 2023, doi:10.3390/ijms24032068_

Round 1
Reviewer 1 Report
The manuscript is well-written and shows relevance. Before being accepted for publication, there are a few points to be fixed...
Line 93. What means MS? It is the first appearance, and then, some explanation should come together with it.
Table 2. please format the font size.
Figure 4. The digits are big and may cause some confusion, please reduce.
Table 4. "Caculated" ??
Figure 7 and 8: Please enlarge them.
Where is the "Conclusion section"? What is the main novelty and advance related to the state of the art?
Author Response
Line 93. What means MS? It is the first appearance, and then, some explanation should come together with it.
-I added: mass spectrometry (MS) analysis
Table 2. please format the font size.
-It is done
Figure 4. The digits are big and may cause some confusion, please reduce.
-I reduced the digits
Table 4. "Caculated" ??
-I corrected
Figure 7 and 8: Please enlarge them. Рисунок 7 и 8:
-It is done
Where is the "Conclusion section"? What is the main novelty and advance related to the state of the art?
- The Discussion section has been replaced with a Conclusion section. The novelty is the study of individual stages of the transketolase reaction, which turned out to be possible only by mass spectrometry and only when carrying out a single-substrate reaction, the rate of which is 50 times lower than the rate of a two-substrate reaction.
Reviewer 2 Report
This paper has no any significant important to the research community. It is unable to attract to readers at large scale and it is suitable only for biochemist. It is better to correspond it some other appropriate journals.
Author Response
Basically you are right, but in this particular case my article was written for publication in a special issue “The Mechanism and Emerging Materials in Thiamine Catalysis” and corresponds to a given topic. The results obtained are of interest for studying the mechanism of catalysis of all thiamine enzymes by the proposed method of mass spectrometry.
Reviewer 3 Report
In this article the author presents a study on the catalytic mechanism of the one-substrate transketolase reaction, by means of mass spectrometry, showing that the reaction requires the participation of water.
In my opinion, the topic of the article is interesting, although the drafting of the article is not optimal.
English should be improved.
The abstract is unclear, the classic scheme of drafting an abstract should be applied, which contains a brief introduction, methods, main results and conclusions in a few lines.
Methods should be described in more detail; for example, in some preparations the temperature is missing.
The discussion paragraph is not a discussion, rather it looks like a conclusion.
The literary survey should be updated. The articles cited were on average published more than ten years ago. Perhaps interest in this topic has waned in recent years?
Author Response
English should be improved.
-I made edits
The abstract is unclear, the classic scheme of drafting an abstract should be applied, which contains a brief introduction, methods, main results and conclusions in a few lines. .
-- I didn't understand a bit. I think it's all there
-brief introduction:
Transketolase catalyzes the interconversion of keto and aldo sugars. Its coenzyme is thiamine diphosphate. Binding of keto sugar with thiamine diphosphate is possible only after C2 deprotonation of its thiazole ring. It was believed that deprotonation occurs due to the direct transfer of a proton to the amino group of its aminopyrimidine ring.
-methods:
Using mass spectrometry,
-main results:
it was shown that a water molecule is directly involved in the deprotonation process. After the binding of thiamine diphosphate with transketolase and its subsequent cleavage, a thiamine diphosphate molecule is formed with a mass increased by one oxygen molecule. After fragmentation, a thiamine diphosphate molecule is formed with a mass reduced by 1 and 2 hydrogen atoms, that is, HO and H2O are split off.
conclusions:
Based on these data, it is assumed that after the formation of holotransketolase, water is covalently bound to thiamine diphosphate, and carbanion is formed as a result of its elimination. This may be a common mechanism for other thiamine enzymes.
-more main results:
The participation of a water molecule in the catalysis of the one-substrate transketolase reaction and a possible reason for the effect of the acceptor substrate on the affinity of the donor substrate for active sites was also shown.
Methods should be described in more detail; for example, in some preparations the temperature is missing.
- I made some additions.
The discussion paragraph is not a discussion, rather it looks like a conclusion. Абзац обсуждения не является обсуждением, скорее он выглядит как заключение.
- I entered a section Conclusion
The literary survey should be updated. The articles cited were on average published more than ten years ago. Perhaps interest in this topic has waned in recent years?
- In recent years, no new results have been obtained related to the research topic. Therefore, I refer to the original articles.Recent researches are of an applied nature or not related to my topic.
Round 2
Reviewer 2 Report
Can be accept in present form